# Preoperative Predictors of Recurrent Tricuspid Regurgitation After Annuloplasty: Insights into the Role of 3D Echocardiography

**DOI:** 10.3390/diagnostics14222515

**Published:** 2024-11-10

**Authors:** Aušra Krivickienė, Dovydas Verikas, Lina Padervinskienė, Vaida Mizarienė, Adakrius Siudikas, Povilas Jakuška, Jolanta Justina Vaškelytė, Eglė Ereminienė

**Affiliations:** 1Department of Cardiology, Medical Academy, Lithuanian University of Health Sciences, LT-44307 Kaunas, Lithuania; 2Institute of Cardiology, Lithuanian University of Health Sciences, LT-50162 Kaunas, Lithuania; 3Department of Radiology, Medical Academy, Lithuanian University of Health Sciences, LT-44307 Kaunas, Lithuania; 4Department of Cardiac, Thoracic and Vascular Surgery, Medical Academy, Lithuanian University of Health Sciences, LT-44307 Kaunas, Lithuania

**Keywords:** tricuspid valve, tricuspid regurgitation, 3D-echocardiography, recurrent tricuspid regurgitation, tricuspid annuloplasty

## Abstract

Background: While tricuspid annuloplasty (TAP) is an effective treatment option for tricuspid regurgitation (TR), understanding the echocardiographic factors contributing to recurrent TR can help in developing more effective preventive measures to reduce the rate of recurrent TR after TAP. Methods: This study was designed as a prospective observational cohort study to investigate factors contributing to recurrent TR following surgical tricuspid valve (TV) repair in patients with moderate or severe functional TR caused by left heart valvular disease, with severe mitral regurgitation as the dominant pathology. The study included 66 patients who underwent preoperative two-dimensional (2D) and three-dimensional (3D) echocardiographic assessments. Patients were divided into two groups based on TAP outcomes: the effective TAP group and the recurrent TR group. Results: The analysis revealed that 3D-derived both septal–lateral diastolic and systolic tricuspid annulus (TA) diameter (odds ratio (OR) 1.77; 95% confidence interval (CI) 1.17–2.68 and OR 1.62; 95% CI 1.14–2.29, respectively), and major axis diastolic TA diameter (OR 1.59; 95% CI 1.15–2.2) had the highest OR among all echocardiographic parameters. The further univariate analysis of predefined echocardiographic values unveiled that the combined effect of heightened 3D-measured TA major axis diastolic diameter and increased right ventricle (RV) basal diameter exhibited the highest OR at 12.8 (95% CI 2.3–72.8) for a recurrent TR. Using ROC analysis, diastolic major axis (area under the curve (AUC) 0.848; cut-off 48.5 mm), septal-lateral systolic (AUC 0.840; cut-off 43.5 mm) and diastolic (AUC 0.840; cut-off 46.5 mm) TA diameter demonstrated the highest predictive value for recurrent TR from all TV parameters. Conclusions: Recurrent moderate or severe TR after TAP is associated with preoperative TA size, right atrium and RV geometry, but not with changes of RV function. The predictive capacity of 2D-assessed echocardiographic parameters was found to be lower when compared to their corresponding 3D parameters.

## 1. Introduction

Secondary tricuspid regurgitation (TR), known as functional tricuspid regurgitation (FTR), is a common tricuspid dysfunction mainly caused by left heart diseases and is a significant clinical challenge that has gained increased attention in recent years. The etiology of TR is secondary in ≥90% of cases due to impaired valve coaptation caused by dilatation of the right ventricle (RV) and/or of the tricuspid annulus (TA).

The prevalence of TR is 15% overall, and it is the most frequent complication of mitral disease [1]. In instances of heart failure (HF), a prevalence of moderate to severe FTR has been documented at an overall rate of 35%. Specifically, severe mitral regurgitation is associated with a 30% incidence, while patients undergoing mitral valve surgery exhibit a higher prevalence of 50% [1,2]. It has been proven in many studies that up to 74% of patients who undergo successful left-sided valve surgery will develop FTR over time [3,4,5]. Late severe TR often leads to poor outcomes and high mortality [2,6,7,8,9].

Tricuspid annuloplasty (TAP), as a safe and effective surgical procedure, is now recommended for the treatment of FTR at the time of left heart surgery [10]. There is increasing concern regarding the residual/recurrent TR after TAP in patients with FTR, with an incidence ranging from 10% to 30% [11,12,13,14,15,16]. Unfortunately, the mechanism and determinants of recurrent TR after tricuspid valve (TV) repair have not been fully investigated and determined. Therefore, it is particularly important to accurately assess the preoperative severity of TR and predict postoperative outcomes.

TA dilatation is known to be a preoperative predictor of recurrent TR. However, there is no consensus on other potential predictors, including the presence of right ventricular HF, pulmonary hypertension, marked RV remodeling/dysfunction, or other TV geometrical alterations. The TA, due to its complex three-dimensional (3D) form, presents several difficulties in its evaluation; however, the use of 3D analysis of the TA remains relatively uncommon. While TAP is an effective treatment option, understanding the echocardiographic factors contributing to recurrent TR can help in developing more effective preventive measures to reduce the rate of recurrent TR after TV repair.

The purpose of this study is to investigate the relationship between TV, right heart geometry, function parameters, and recurrent TR after TAP.

## 2. Materials and Methods

### 2.1. Study Population and Study Design

This study was designed as a prospective observational cohort study to investigate factors contributing to recurrent TR following surgical TV repair in patients with moderate or severe FTR caused by left heart valvular disease.

From July 2018 to December 2021, 67 patients were diagnosed with moderate-severe functional TR associated with left heart valvular disease and underwent concomitant TAP during left heart surgery. TV repair (annuloplasty) was always performed in patients with moderate-to-severe or severe TR. Indication of TV repair was followed according to the 2021 ESC Guidelines [10]. TV repair using suture annuloplasty by De Vega was the technique of choice in all patients. Baseline characteristics, preoperative 2D and 3D echocardiographic data, and clinical outcomes were evaluated for all patients.

Patients with primary TR, ischemic heart disease (assessed by coronary angiography), infective endocarditis, chronic pulmonary disease, congenital heart disease, or other causes of precapillary pulmonary hypertension were excluded from the study. Patients who underwent re-do TV repair were also excluded (detailed inclusion and exclusion criteria are presented in Appendix A).

Surgical procedures of the left heart included the following: mitral valve repair, mitral valve replacement, aortic valve replacement, mitral valve repair combined with aortic valve replacement, and mitral valve replacement combined with aortic valve replacement.

The follow-up period is 15 months (mean 15.57 ± 4.19 months). All patients were followed up with transthoracic 2-dimensional (2D) and 3D echocardiography ≥ 1-year after TAP. The primary endpoint was the recurrence of significant TR (defined as moderate and severe regurgitation).

We divided the patients into two groups based on the outcomes of TAP: effective TAP and recurrent TR group. Effective TAP was defined as mild TR (≤2/4 grade) 1-year after surgery, while recurrent TR group was defined as moderate, moderate-severe, or severe TR (>2/4 grade).

The study was approved by the Kaunas Regional Biomedical Research Ethics Committee, No. BE-2-64, issued on 24 July 2018. All patients provided informed consent.

### 2.2. Transthoracic Echocardiography

The 2D and 3D preoperative transthoracic echocardiography was performed using a GE VingMed VividE95 (GE Vingmed Ultrasound AS, Horten, Norway) imaging system, equipped with a M3S 4.0 MHz transducer, capable of displaying 3D images. An experienced independent echocardiographer blinded to the patient’s clinical data performed the echocardiographic studies. Digital loops were stored and analyzed offline (GE Vingmed, Phillips TomTec, Unterschleissheim, Germany). The 3D TV analysis was made using the 4D Auto TVQ quantification software package (EchoPac v204, GE Healthcare, Horten, Norway).

Anatomic and Doppler examinations and measurements were performed according to recent American Society of Echocardiography recommendations and European Association of Cardiovascular Imaging (EACVI) guidelines [17,18]. The 2D and 3D echocardiography was performed and included the following parameters: the left ventricular (LV) and left atrial (LA) geometry (diameters and volumes), LV ejection fraction, the RV geometry (RV diameters, areas and volumes), functional (velocity of the tricuspid annular systolic motion (S′), tricuspid annulus plane systolic excursion (TAPSE), RV fractional area change (FAC), RV ejection fraction (RVEF), and strain (RV free wall longitudinal strain (RV FWLS) and RV septal longitudinal strain (RV SLS)), the tricuspid valve 2D (the systolic and diastolic four-chambers diameters), the leaflet tenting height (the distance between the coaptation of the septal and anterior leaflets and the TA plane) and area (the area enclosed by the annular plane and septal and anterior leaflets), and 3D (the systolic and diastolic four-chambers (septal–lateral), two-chambers (anterior–posterior), major axis TA diameters, TA area, perimeter, leaflet tenting height and volume (the volume enclosed between the plane of the TA and TV leaflets)), right atrium (RA) (diameter, length, area and volume) parameters. Parameters have been indexed to body surface area.

Volume datasets were obtained under breath-hold to avoid stitch artifacts using a multi-beat full-volume model in the four-chamber apical view focused on the RV and multi-beat 3D zoom mode in the four-chamber apical view focused on the TV. The views were optimized for depth and gain setting before 3D acquisition and close attention was given to including the entire TV or RV in the sector boundaries. A multi-slice display was used during acquisition to ensure a complete inclusion of the RV in the dataset [19]. The RV and TV multi-beat 3D views dataset was also acquired with the narrowest possible depth under breath-hold to obtain a higher volume rate (higher than 20 volumes per second).

The digitally stored multi-beat full volume dataset in the apical axis four-chamber view was imported into the dedicated software tools (4D RV-Analysis 2.0, TomTec Imaging Systems, Unterschleissheim, Germany) to calculate RV end-diastolic and end-systolic volumes and RVEF.

The digitally stored live 3D zoom dataset was imported into the 4D Auto TVQ quantification (GE Healthcare, Horten, Norway) workstation to analyze the TV geometry (Figure 1). Anterior–posterior and septal–lateral diameters of the TV, as well as major and minor axis diameter, the TV annular area, perimeter, and sphericity index, were obtained on a mid-systolic and mid-diastolic frame. The closed leaflets were traced in mid-systole on successive equidistant long-axis planes to obtain the leaflet tethering height and 3D tenting volume.

The severity of TR was measured quantitatively according to the recent ESC guidelines [20], using TR effective regurgitant orifice area (EROA) (PISA) and biplane TR vena contracta (VC) (the width of the color jet at its narrowest point) from the 2D apical four-chamber view. According to a recently proposed grading scheme for TR, patients with biplane VC 3–6.9 mm and TR EROA 20–39 mm^2^ were considered moderate, and VC ≥ 7 mm and EROA ≥ 40 mm^2^ were considered severe TR [20,21,22].

### 2.3. Statistical Analysis

Continuous variables were represented as median values with interquartile ranges (IQR) and compared using non-parametric Wilcoxon rank-sum tests. Categorical variables were presented as counts and percentages. Prior to analysis, all variables underwent normality testing using the Shapiro–Wilk test. Univariate and multivariate logistic regression analyses were conducted to identify associations between right heart and TV parameters and moderate-severe recurrent FTR. Variables demonstrating significance in the univariate analysis were subsequently included in the multivariable-adjusted analysis to elucidate their collective predictive capacity for the recurrent TR. Results were expressed as odds ratios (ORs) with corresponding 95% confidence intervals (CIs). The optimal model was selected by balancing the inclusion of factors with the accuracy of logistic regression analysis. Receiver operating characteristic (ROC) analysis was performed to determine the optimal cut-off values for 2D and 3D echocardiographic parameters in predicting recurrent TR. The accuracy of these cut-off values was assessed using the area under the ROC curve (AUC). Statistical analyses were carried out using SPSS version 27 (IBM, Armonk, NY, USA), and *p*-values < 0.05 were deemed statistically significant.

## 3. Results

Among the study cohort, 54% were male, and the mean age was 68 ± 9 years. Following surgery, one patient experienced a fatal outcome due to pneumonia after a 4-day postoperative period and was subsequently excluded from the study. The remaining patients were subject to ongoing follow-up, during which no instances of mortality were observed. The mean duration of the follow-up period was 15.6 ± 4.2 months. There were 53 patients (80%) diagnosed with moderate FTR and 13 patients (18%) with severe FTR. There was no recurrence of moderate or severe mitral or aortic regurgitation and stenosis. None of the patients had a postoperative pacemaker implanted. Out of the patients monitored for more than 1 year, 16 individuals (24.2%) experienced a recurrence of moderate or severe TR. The remaining 50 patients (75.8%) did not exhibit significant (>2/4 grade) recurrent TR following TAP.

### 3.1. Analysis of Risk Factors of Recurrent Tricuspid Regurgitation

The distribution of baseline characteristics and echocardiographic parameters among the study groups are shown in Table 1 and Appendix A.

Analysis revealed that age, gender, body mass index, preoperative transthoracic echocardiography-based left ventricular and atrial geometrical and functional parameters or pulmonary artery pressure did not differ between effective TAP and recurrent TR groups (Appendix A). Although preoperative effective regurgitant TV orifice area did not differ between the groups (recurrent TR vs. effective TAP: 38.37 [32.69] mm^2^ vs. 29.45 [14.9] mm^2^, *p* = 0.117, respectively), the following differences of right heart parameters were identified between these two groups and are detailed in Table 1. Recurrent TR was associated with larger RV and RA geometrical and volumetrical parameters; however, RV functional parameters did not show significant associations.

Table 2 shows the differences in TA geometry between the two groups: the septal–lateral and major axis TA diameters, as well as TA area and perimeter, were increased in the recurrent TR group when compared to the effective TAP group. Even though the TA sphericity and TV leaflet tethering did not differ between the groups, recurrent TR was associated with larger tenting area and volume.

### 3.2. Relation of TV and RV Geometry and Recurrent Tricuspid Regurgitation

In order to evaluate the influence of TV geometry and right-heart remodeling on recurrent TR, a regression analysis was conducted. Univariate analysis was executed, and ORs are presented in Table 3 and Appendix A. None of the left-heart parameters exhibited predictive value. Conversely, RA parameters demonstrated predictive value for recurrent regurgitation after TAP. Among all RA parameters, RA diameter exhibited the highest odds ratio (OR 1.13; 95% CI 1.03–1.23). However, RV parameters demonstrated even greater predictive value. RV basal diameter index (OR 1.34; 95% CI 1.09–1.65), RV end-systolic area index (OR 1.26; 95% CI 1.02–1.56), and RV parasternal diastolic diameter (OR 1.22; 95% CI 1.06–1.40) exhibited the highest predictive value among all RV parameters for recurrent regurgitation.

Despite the high odds ratios for RV parameters, further analysis revealed that TA parameters, both 2D and 3D, possessed the highest predictive value (Table 3). Specifically, both septal–lateral diastolic and systolic TA and major axis diastolic diameters demonstrated the highest prognostic value among 3D parameters. In the analysis of 2D TA parameters, the four-chamber systolic diameter exhibited the highest predictive value (OR 1.44; 95% CI 1.14–1.83). To mitigate potential confounding factors related to age and gender, model I analysis was conducted. The results indicated an increase in OR for the previously mentioned parameters, while no significance was lost.

To further assess the diagnostic efficacy of 2D and 3D echocardiographic right-heart parameters, univariate analysis of predefined echocardiographic values was conducted. Utilizing ROC analysis, parameters with the highest predictive value were identified: major axis diastolic TA diameter (AUC 0.848; cut-off 48.5 mm), TV leaflet tenting volume (AUC 0.778; cut-off 5.1 mL), and RV basal diameter (AUC 0.763; cut-off 47.5 mm). The interrelation of these cut-off values with recurrent regurgitation is revealed in Table 4. The analysis unveiled that the combined effect of increased RV basal diameter and heightened TA major axis diastolic diameter derived from 3D analysis exhibited the highest OR at 12.8 (95% CI 2.3–72.8). This value further increased upon the elimination of age and gender as potential confounding factors.

Multivariate stepwise regression analysis was conducted, and the results are presented in Table 5. Notably, among all parameters considered, only FAC, RV basal diameter, and septal–lateral systolic TA diameter retained statistical significance. Intriguingly, during the multivariate analysis, an increase in septal–lateral systolic TA diameter exhibited a lower OR, while the predictive value trend for RV basal diameter persisted. Additionally, the multivariate analysis identified a new parameter: FAC, whose higher value was associated with a lower OR for recurrent regurgitation.

### 3.3. Prediction of Recurrent Tricuspid Regurgitation

The ROC analysis of selected echocardiographic parameters, displaying the highest predictive value in this patient cohort, is presented in Table 6 and Appendix A. Among the analyzed RV parameters, RV basal diameter (AUC 0.763; cut-off 44.5 mm) and RV parasternal diastolic diameter (AUC 0.747; cut-off 34.5 mm) exhibited the most robust predictive value for recurrent TR. Regarding TV parameters, the diastolic major axis (AUC 0.848; cut-off 48.5 mm), as well as septal–lateral systolic or diastolic TA diameter (both AUC 0.840; cut-off 43.5 mm and 46.5 mm, respectively), demonstrated the highest predictive value in this particular group of patients (Figure 2).

Conversely, the predictive capacity of 2D-assessed echocardiographic parameters was found to be lower when compared to their corresponding 3D parameters. Specifically, the comparison between the four-chamber systolic diameter (2D) and septal–lateral systolic TA diameter (3D) yielded AUC values of 0.799 and 0.840, respectively. Similarly, the comparison between TV tenting area (2D) and TV leaflet tenting volume (3D) resulted in AUC values of 0.711 and 0.778, respectively.

## 4. Discussion

In this prospective study, we analyzed the preoperative echocardiographic 2D and 3D data of 66 individuals with moderate or severe FTR due to left-heart valvular disease who underwent concomitant TAP during left heart surgery. The main insights gained from our study can be summarized as follows: (a) the recurrence of moderate-severe FTR following TAP is linked to the geometry of the RA and RV before surgery; (b) no correlation was identified between the recurrence of TR and alterations in LV and RV function; (c) parameters with the highest predictive value for recurrent TR were 3D-derived major axis diastolic and systolic TA diameter, both septal–lateral systolic and diastolic TA diameter, TA perimeter and TV tenting volume; (d) the predictive capacity of 2D-assessed echocardiographic parameters was found to be lower when compared to their corresponding 3D parameter.

### 4.1. The Association Between the RV and LV Geometrical and Functional Parameters and Recurrent TR

Kitamura et al., using multivariate analysis, showed that a larger RV systolic diameter was a predictor of recurrent ≥ 2 + TR [23]. Kitamura’s study demonstrated that both RV diastolic and systolic diameters were significantly larger in the recurrence TR group compared to the non-recurrence group, and no differences were observed between the two groups in RV FAC. Calafiore et al. reported that RV dilatation was a risk factor of recurrent TR after TAP, including the result of TV repair with a classic ring or flexible ring [24]. Maslow et al. reported that for patients with FTR undergoing primary left heart surgery, preoperative RV width and the predictive value of RV width > 4.88 cm were associated with TV repair failure [25]. RV dilatation causes free wall dilatation and extends the length between papillary muscles [26], resulting in TR. Our study showed that comparing recurrent TR and effective TAP groups, recurrent TR was associated with larger RV and RA geometrical and volumetrical parameters; however, RV functional parameters did not differ between these two groups. Using ROC analysis, RV basal diameter showed the most reliable predictive value for recurrent TR from RV parameters and was identified as one of the three parameters with the highest predictive value for TR recurrence in our study. Multivariate stepwise regression analysis confirmed that RV basal diameter was an independent predictor of recurrent TR.

It is known that recurrent TR is associated with severe LV dysfunction [27,28]. LV dysfunction causes pressure and/or volume overload, resulting in LV dilatation. In addition, elliptical change in shape is frequently encountered in patients with severe LV dysfunction. These geometric LV abnormalities affect RV geometry and function directly through the interventricular septum or indirectly contributing to pulmonary hypertension [28]. Gatti and colleagues showed that LVEF < 50%, whereas Kitamura et al. reported that LVEF < 40% was a predictor of grade ≥ 2 + recurrent TR after device annuloplasty [23,27]. Although, in our study, LV EF was also lower in the recurrent TR group, there was no significant association between LV function and recurrence of TR. This may have been due to the fact that there were only a few patients with severe LV dysfunction in our study group, while in Kitamura and colleagues’ study, even 32% of patients had LV EF < 40% in the recurrence TR group.

### 4.2. The Association Between TV Geometrical Parameters and Recurrent TR

In adults, the normal TA diameter is 28 ± 5 mm (four-chamber view in diastole). Current ESC/EACTS guidelines suggest a threshold diameter ≥ 40 mm (>21 mm/m^2^) as an indicator for surgery [10]. Several echocardiographic studies have suggested important changes in the TV geometry in patients with FTR, including annular dilatation and tethering of the leaflets [29,30,31]. These TV deformations may restrict the motion of the leaflets and decrease coaptation.

Previous studies showed TA size is a risk factor for recurrent TR in TV repair with 3D-shaped rings [32,33]. Several studies also have been conducted with the aim of exploring the association between TA circumference and postoperative TR and as a result, there is a close association between the circumference and TR. Jin Guo Xu et al. found that TA circumference index ≥ 7.86 cm/m^2^ is the risk factor of recurrent significant TR after concomitant TAP during left heart surgery [34]. Zhu et al. have recommended that a TA circumference index of 83 mm/m^2^ is the threshold of prophylactic TAP [35]. Based on Mohammad Sharif Popal’s recommendation, the threshold is set as 80.2 mm/m^2^ [36]. Our study also confirmed the high predictive value of TA perimeter and TA perimeter index for TR recurrence (AUC 0.790; cut-off 130.5 mm and AUC 0.780, cut-off 7.8 cm/m^2^, respectively). However, analysis revealed that 3D-derived TA diameters showed even higher predictive value for TR recurrence. Using ROC analysis, the diastolic major axis (cut-off 48.5 mm), as well as septal–lateral systolic (cut-off 43.5 mm) and diastolic (cut-off 46.5 mm) TA diameters demonstrated the highest predictive value for recurrent TR from all TA parameters. The further univariate analysis of predefined echocardiographic values unveiled that the combined effect of heightened 3D-measured TA major axis diastolic diameter (>48.5 mm) and increased RV basal diameter (>47.5 mm) exhibited the highest OR at 12.8 (95% CI 2.3–72.8) for a recurrent TA. Similar prognostic importance of diameters was seen in other studies. Maslow and colleagues reported that for patients with FTR undergoing primary left heart surgery, preoperative TA diameter was a predictor of repair outcome [25]. They also found prognostic values: preoperative and intraoperative TA diastolic diameter ≥ 4.2 cm and preoperative systolic TA diameter ≥ 3.7 cm predicted repair failure. Dilated TA may accompany severe damage of the TV lesion, which is attributed to the presence of recurrent regurgitation. Once TA is dilated, RV remodeling will occur. After TAP, original TV mismatches altered annulus due to heart remodeling, which leads to possible regurgitation. Thus, the dilatation of TA plays an important role in the recurrence of FTR, and furthermore, as our study showed, septal–lateral systolic TA diameter is an independent predictor of postoperative recurrent TR.

Increased apical displacement of the tricuspid leaflets (tethering) is evaluated by 2D-echocardiography by measuring the tenting area and coaptation distance in the four-chamber view. If the tethering distance is >8 mm and the tethering area is >1.6 cm^2^, it can be defined that the tethering extent of tricuspid leaflets is significant [37]. Using 3D echocardiography, we can evaluate another tethering parameter-tenting volume. Fukuda and colleagues showed that preoperative leaflet tethering height and area predicted early and mid-term (>1 year) outcomes of annuloplasty [12,28]. Our study demonstrated that recurrent TR was associated with larger tenting area and volume; TV tenting area > 127 mm^2^ and TV tenting volume > 5.1 mL were the risk factors of recurrent TR after concomitant TAP during left heart valvular surgery. The univariate analysis of predefined echocardiographic values unveiled that the combined effect of TV tenting volume of more than 5.1 mL and TA major axis diastolic diameter of more than 48.5 mm displayed the OR at 8.5 (95% CI 1.7–41.5). Chen suggests that 3D-derived TV tethering volume is an important preoperative parameter associated with adverse events (occurrence of heart failure requiring hospital admission or all-cause mortality) in patients undergoing TAP, and the ≥3.9 mL tethering volume provides important predictive value at 1-year follow-up [38]. Including our result, these reports indicate the limitation of suture annuloplasty by De Vega for severe tethered TV. Current suture annuloplasty techniques might not be adequate to correct TR in patients with severe tethering, and an additional procedure (e.g., repair with ring, leaflet augmentation, clover technique, right ventricular papillary muscle approximation) may be required [39,40,41,42].

### 4.3. The Importance of 3D Analysis for TV Evaluation

Another question that arises is whether 3D analysis is important in TV evaluation. TA is a complex three-dimensional structure, and the most suitable echocardiographic technique to obtain accurate measurements of TA size is 3D echocardiography [43,44,45]. Moreover, the size of the TA is critical in determining the need for concomitant TV interventions in patients undergoing left-sided valve surgery. Two-dimensional echocardiographic evaluation of the TA diameter underestimates the actual size due to its elliptical shape. Consequently, the annular diameter measured in a four-chamber (septal–lateral) view or a short-axis view (oblique) is less than the true anteroposterior diameter [46]. Moreover, when the TA dilates, it does so mostly in the anteroposterior direction, a direction that is not explored in the conventional 2D echocardiography apical four-chamber view [47]. The advent of 3D echocardiography enables detailed evaluation of the TV apparatus; nonetheless, the clinical value of preoperative 3D echocardiography is unknown in patients undergoing TAP. In our study, we used 3D echocardiography, and TA parameters were measured by the multiplanar reconstruction method, which allows for a much more accurate assessment of TV parameters. Our study demonstrated that the predictive capacity of 2D-assessed echocardiographic parameters was found to be lower when compared to their corresponding 3D parameters (four-chamber systolic diameter (2D) and septal–lateral systolic TA diameter (3D) yielded AUC values of 0.799 and 0.840; and TV tenting area (2D) and TV tenting volume (3D) resulted in AUC values of 0.711 and 0.778, respectively).

### 4.4. Study Limitations

The limitations of our study can be described as follows: first, one of the main limitations of this study was that a single tricuspid valve repair technique (suture annuloplasty) was used, which limits generalizability to other forms of tricuspid valve repair. Second, this was a single-center study with a small number of patients (*n* = 66), and the significance should be tested in a multi-center cohort.

## 5. Conclusions

Recurrent moderate or severe TR after TAP is associated with preoperative TA size, RA, and RV geometry but not with changes in RV function. Key predictors, including 3D echocardiography measured septal–lateral systolic (cut-off value 43.5 mm) and diastolic (cut-off value 46.5 mm) TA diameters, major axis diastolic TA diameter (cut-off value 52.5 mm), TA perimeter (cut-off value 130.5 mm), TV leaflet tenting volume (cut-off value 5.1 mL), and RV basal diameter (cut-off value 47.5 mm) were identified, emphasizing their significance in predicting TR recurrence. The combination of predefined RV basal diameter and major axis diastolic TA diameter showed the highest odds ratio for recurrent TR. This information can guide surgical decision-making, such as considering earlier or more extensive TV repair procedures or the need for close postoperative monitoring in patients at higher risk of recurrence in order to improve patient outcomes and reduce the need for reoperations.

## Figures and Tables

**Figure 1 diagnostics-14-02515-f001:**
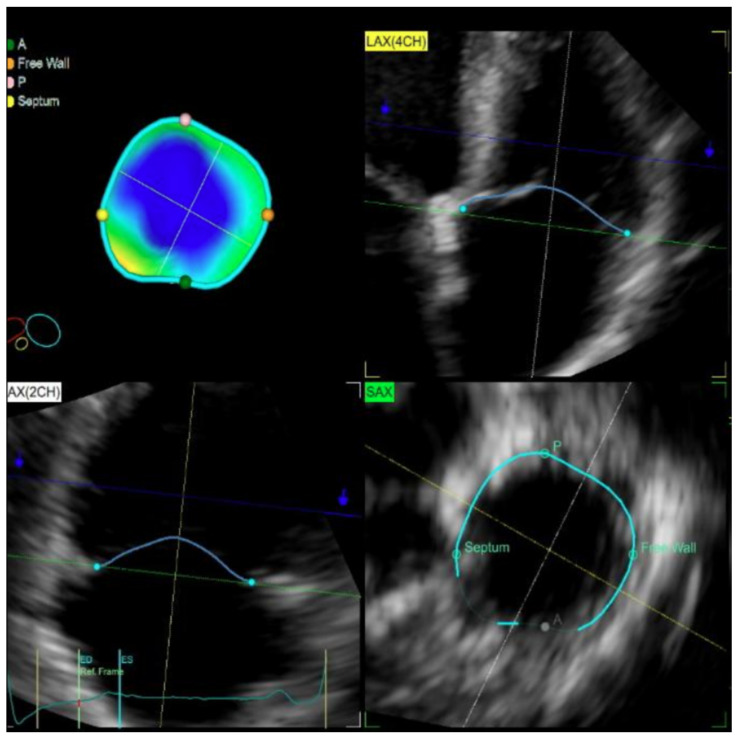
Three-dimensional TV echocardiographic measurements.

**Figure 2 diagnostics-14-02515-f002:**
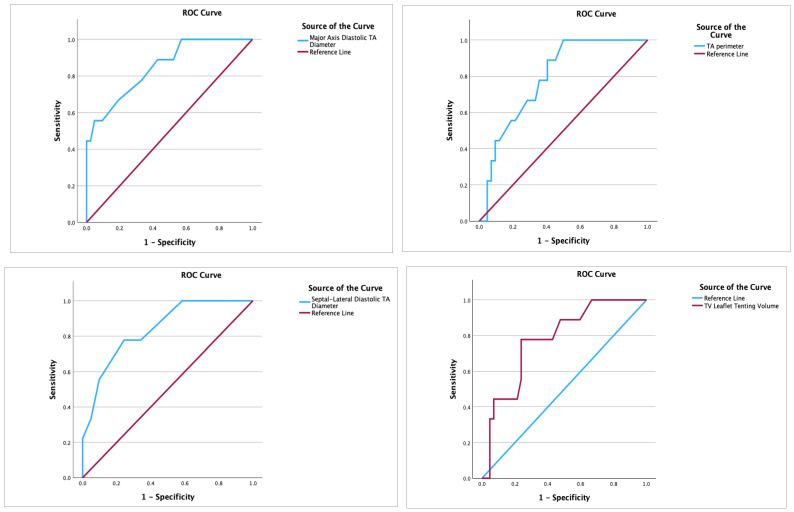
ROC analysis for prediction of recurrent TR.

**Table 1 diagnostics-14-02515-t001:** Two-dimensional and three-dimensional right-heart echocardiographic characteristics of study population.

	Effective TAP Group *n* = 50 (75.8%)	Recurrent TR Group *n* = 16 (24.2%)	*p*-Value
**RA parameters**			
Diameter, mm	49 (9.3)	54 (7.8)	**0.005**
Diameter index, mm/m^2^	24.8 (5.7)	29.5 (5.0)	**0.005**
Length, mm	60.5 (11.3)	66.4 (7.6)	**0.003**
Area, cm^2^	26.9 (10.6)	30.4 (11.5)	**0.008**
Area index, cm^2^/mm^2^	14.2 (5.2)	17.2 (6.9)	**0.002**
Volume, mL	92 (64)	106 (64)	**0.026**
**RV parameters**			
Parasternal diastolic diameter, mm	35 (7)	38.6 (6.1)	**0.003**
Parasternal systolic diameter, mm	27 (6.5)	30.5 (7.3)	**0.003**
Basal diameter, mm	44.5 (7)	51 (11.3)	**0.002**
Basal diameter index, mm/m^2^	23.3 (3.6)	26.8 (5.5)	**0.006**
Middle diameter, mm	35 (7.3)	39 (8.5)	**0.012**
Middle diameter index, mm/m^2^	17.8 (4.1)	21.5 (6.8)	**0.027**
Length, mm	63 (18.3)	64 (15)	0.736
Sphericity index, %	0.55 (0.13)	0.61 (0.26)	**0.041**
End-diastolic area, cm^2^	21.2 (9.5)	22.7 (10.6)	**0.046**
End-diastolic area index, cm^2^/m^2^	10.8 (4.5)	14.2 (4.7)	**0.017**
End-systolic area, cm^2^	13.9 (7.2)	16.3 (10.1)	0.067
End-systolic area index, cm^2^/m^2^	6.8 (3.5)	9 (4)	**0.017**
End-diastolic volume, mL	147 (90)	221 (137)	0.220
End-systolic volume, mL	87 (66)	130 (98)	0.176
RV EF, %	41 (7)	38 (7)	0.211
FAC, %	33 (10)	30 (10)	0.231
TAPSE, mm	18 (9)	17 (4)	0.346
S′, cm/s	10.4 (5)	9.7 (4.8)	0.646
Septal wall strain, %	−11.1 (5.1)	−9.8 (4.8)	0.373
Lateral wall strain, %	−20.4 (6.1)	−18.4 (11.9)	0.673

EF—ejection fraction, RV—right ventricle, FAC—fractional area change, TAPSE—tricuspid annular plane systolic excursion, S′—tricuspid lateral annular systolic velocity, RA—right atrium, TR—tricuspid regurgitation, TAP—tricuspid annuloplasty.

**Table 2 diagnostics-14-02515-t002:** Two-dimensional and three-dimensional TV echocardiographic characteristics of study population.

	Effective TAP Group *n* = 50 (75.8%)	Recurrent TR Group *n* = 16 (24.2%)	*p*-Value
**TV 2D parameters**			
TA diastolic diameter, mm	41.5 (4)	45.4 (5)	**<0.001**
TA diastolic diameter index, mm/m^2^	21.8 (2.8)	23.9 (6.5)	**0.012**
TA systolic diameter, mm	39 (5.5)	42 (5.6)	**0.003**
TA systolic diameter index, mm/m^2^	19.9 (2.4)	22.3 (6.5)	**0.026**
TV leaflet tethering height, mm	5.3 (2.1)	6.6 (2.4)	0.073
TV tenting area (mm^2^)	120 (66)	170 (96)	**0.012**
TV EROA, mm^2^	29.45 (14.9)	38.37 (32.69)	0.117
**TV 3D parameters**			
TA area, cm^2^	14.2 (4.0)	16.7 (2.8)	**0.010**
TA area index (cm^2^/m^2^)	6.96 (2.08)	8.79 (2.38)	**0.004**
TA perimeter, mm	127 (22)	143 (18)	**0.005**
TA perimeter index (cm/m^2^)	6.47 (1.35)	7.83 (1.81)	**0.007**
Septal–Lateral Systolic TA Diameter, mm	42 (4)	46 (5)	**<0.001**
Septal–Lateral Systolic TA Diameter Index, cm/m^2^	2.14 (0.31)	2.37 (0.81)	**0.04**
Septal–Lateral Diastolic TA Diameter, mm	44 (5)	48 (5)	**<0.001**
Septal–Lateral Diastolic TA Diameter Index, cm/m^2^	2.26 (0.37)	2.63 (0.87)	**0.04**
Anterior–Posterior TA Diameter, mm	40 (7)	44 (6)	0.081
Anterior–Posterior TA Diameter Index, cm/m^2^	2 (0.48)	2.22 (0.57)	**0.045**
Major Axis Systolic TA Diameter, mm	46 (5)	49 (7)	**0.001**
Major Axis Systolic TA Diameter Index, cm/m^2^	2.25 (0.34)	2.68 (0.73)	**0.003**
Major Axis Diastolic TA Diameter, mm	48 (6)	53 (6)	**<0.001**
Major Axis Diastolic TA Diameter Index, cm/m^2^	2.36 (0.34)	2.79 (0.81)	**0.007**
TV Leaflet Coaptation point Height, mm	9 (5)	10 (5)	0.653
TV Leaflet Tenting Volume, mL	3.75 (1.9)	5.3 (2.8)	**0.008**
TV Sphericity Index, %	83 (11)	84 (12)	0.397

TV—tricuspid valve, TA—tricuspid annulus, EROA—effective regurgitant orifice area, TAP—tricuspid annuloplasty.

**Table 3 diagnostics-14-02515-t003:** Relation of TV parameters with recurrent TR.

	Univariate			Model I		
	OR	95% CI.	*p*-Value	OR	95% CI.	*p*-Value
**3D echo-derived TV parameters**						
TA perimeter, mm	1.07	1.01–1.13	**0.024**	1.11	1.03–1.21	**0.008**
Septal–Lateral Systolic TA Diameter, mm	1.62	1.14–2.29	**0.007**	1.96	1.22–3.14	**0.005**
Anterior–Posterior TA Diameter, mm	1.17	0.97–1.4	0.097	1.25	0.99–1.57	0.061
Major Axis Systolic TA Diameter, mm	1.51	1.15–1.99	**0.003**	1.75	1.20–2.55	**0.004**
TV Leaflet Tenting Volume, mL	1.42	1.01–1.98	**0.043**	1.41	0.99–1.98	0.052
TA area, cm^2^	1.55	1.06–2.27	**0.025**	2.03	1.16–3.53	**0.013**
Septal–Lateral Diastolic TA Diameter, mm	1.77	1.17–2.68	**0.007**	1.99	1.14–3.46	**0.015**
Major Axis Diastolic TA Diameter, mm	1.59	1.15–2.2	**0.005**	1.75	1.17–2.6	**0.006**
TV Sphericity Index, %	0.92	0.83–1.031	0.155	0.93	0.83–1.05	0.255
**2D echo-derived TV parameters**						
4-Chambers Systolic Diameter, mm	1.44	1.14–1.83	**0.002**	1.67	1.22–2.28	**0.001**
4-Chambers Diastolic Diameter, mm	1.31	1.09–1.58	**0.005**	1.40	1.1–1.76	**0.005**
TV leaflet tethering height, mm	1.31	0.93–1.85	0.121	1.38	0.95–1.99	0.093
TV tenting area (mm^2^)	1.02	1.003–1.03	**0.01**	1.02	1.004–1.03	**0.007**
TV EROA, mm^2^	1.03	0.99–1.07	**0.058**	1.03	0.99–1.07	**0.062**

Model I—univariate regression adjusted to age and gender, TA—tricuspid annulus, TV—tricuspid valve, EROA—effective regurgitant orifice area.

**Table 4 diagnostics-14-02515-t004:** Prognostic value of predefined echocardiographic values of recurrent TR (univariate analysis).

	Univariate			Model I		
	OR	95% CI.	*p*-Value	OR	95% CI.	*p*-Value
Major Axis Diastolic TA Diameter * + TV Leaflet Tenting Volume **	8.5	1.7–41.5	**0.008**	10.4	1.7–65.6	**0.012**
Major Axis Diastolic TA Diameter * + RV basal diameter ***	12.8	2.3–72.8	**0.004**	15.7	2.4–101.7	**0.004**
TV Leaflet Tenting Volume ** + RV basal diameter ***	8.5	1.7–41.5	**0.008**	8.7	1.6–47.5	**0.012**
Major Axis Diastolic TA Diameter * + TV Leaflet Tenting Volume ** + RV basal diameter ***	6.3	1.3–29.3	**0.02**	6.8	1.2–37.4	**0.028**

Model I—univariate regression adjusted to age and gender. * Major Axis Diastolic TA Diameter > 48.5 mm; ** TV Leaflet Tenting Volume > 5.1 mL; *** RV basal diameter > 47.5 mm. RV—right ventricle, TA—tricuspid annulus, TV—tricuspid valve.

**Table 5 diagnostics-14-02515-t005:** Prognostic value of predefined echocardiographic values of recurrent TR (multivariate analysis).

	Multivariate			Model I		
	OR	95% CI.	*p*-Value	OR	95% CI.	*p*-Value
Septal–Lateral Systolic TA Diameter, mm	0.75	0.59–0.99	**0.048**	0.64	0.43–0.96	**0.03**
TV Leaflet Tenting Volume, ml	1.15	0.77–1.71	0.505	1.17	0.77–1.78	0.466
RV basal diameter, mm	1.42	1.04–1.94	**0.027**	1.44	1.02–2.01	**0.036**
FAC, %	0.87	0.76–0.99	**0.035**	0.84	0.71–0.98	**0.031**
RA diameter, mm	0.96	0.79–1.15	0.647	0.99	0.81–1.23	0.976

Model I—multivariate regression adjusted to age and gender. TA—tricuspid annulus, TV—tricuspid valve, RV—right ventricle, FAC—fractional area change, RA—right atrium.

**Table 6 diagnostics-14-02515-t006:** Prediction of recurrent TR.

Variables	AUC	*p*-Value	Cut-Off	Sensitivity	Specificity
**3D echo-derived TV parameters**					
TA perimeter, mm	0.790	**0.007**	130.5	89	60
Septal–Lateral Systolic TA Diameter, cm	0.840	**0.001**	43.5	78	71
Septal–Lateral Diastolic TA Diameter, cm	0.840	**0.002**	46.5	78	76
Anterior–Posterior TA Diameter, cm	0.687	0.082			
Major Axis Systolic TA Diameter, cm	0.833	**0.002**	47.5	78	71
Major Axis Diastolic TA Diameter, cm	0.848	**0.001**	48.5	89	57
TV Leaflet Tenting Volume, mL	0.778	**0.009**	5.1	78	76
TA area, cm^2^	0.770	**0.012**	14.35	89	60
TV Sphericity Index, %	0.592	0.391			
**2D echo-derived TV parameters**					
4-Chambers Systolic Diameter, mm	0.799	**< 0.001**	42.5	81	56
4-Chambers Diastolic Diameter, mm	0.746	**0.003**	38.5	94	46
TV leaflet tethering height, mm	0.650	0.074			
TV tenting area (mm^2^)	0.711	**0.012**	127	75	62
TV EROA, mm^2^	0.632	0.117			

TA—tricuspid annulus, TV—tricuspid valve, EROA—effective regurgitant orifice area.

## Data Availability

The original contributions presented in the study are included in the article/Appendix A, further inquiries can be directed to the corresponding author.

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
