# Peer review of "Preoperative Predictors of Recurrent Tricuspid Regurgitation After Annuloplasty: Insights into the Role of 3D Echocardiography"

_diagnostics, 2024, doi:10.3390/diagnostics14222515_

Round 1

Reviewer 1 Report

Comments and Suggestions for Authors

In the manuscript, the authors present a study regarding the importance of preoperative 3D ecocardiography in predicting recurrent tricuspid regurgitation after annuloplasty. They included in the study 66 patients. The general idea of the manuscript is quite interesting and the conclusions of the manuscript are pertinent. The manuscript can be published, but in order to improve the quality of the manuscript, in my opinion, some changes have to be done. My observations are :

- the authors stated that they included in the manuscript 66 patients, but they also have some exclusion criteria. They did not present the total number of patients that were studied (before the application of the exclusion criteria)

- please introduce the limitations of the study at the end of the discussions part of the main manuscript.

Author Response

We would like to thank you for your comments regarding the manuscript. We addressed the questions you provided:

- the authors stated that they included in the manuscript 66 patients, but they also have some exclusion criteria. They did not present the total number of patients that were studied (before the application of the exclusion criteria)

We enrolled patients in our study using inclusion and exclusion criteria. A total of 67 patients were enrolled, but 1 of them died during follow-up and was subsequently withdrawn from the study, leaving a total of 66 patients. We added a table with summarized inclusion and exclusion criteria in our article (Table S1).

- please introduce the limitations of the study at the end of the discussions part of the main manuscript.

We moved the paragraph "Study limitations" to the end of the "Discussion" part.

Reviewer 2 Report

Comments and Suggestions for Authors

This is an interesting prospective observational cohort study about preoperative predictors of recurrent tricuspid regurgitation after annuloplasty. The discussion and the tables are well done. However, i have some considerations to make:

- In the introduction, the authors could consider to expand the epidemiologial data (the prevalane of secondary tricuspid regurgitation for example);

- Also in the introduction, the acronym tricuspid valve (TV) should be placed first in the text as it has already been mentioned previously in the text (it should be also explained in the abstract, albeit easily interpretable);

- In the methods, it might be useful to summarize the inclusion and exclusion criteria by creating a specific table;

- In the results, the authors must decide to have a single criterion when using numbers , without alternating numbers and letters (es. 68 vs four);

- Minor typos (mostly in spacing, could be a layout error during the visualization though).

Comments on the Quality of English Language

Minor typos (mostly in spacing, could be a layout error during the visualization though).

Author Response

We would like to thank you for your comments regarding the manuscript. We addressed the questions you provided:

- In the introduction, the authors could consider to expand the epidemiologial data (the prevalane of secondary tricuspid regurgitation for example);

We expanded the “Introduction” section with some additional epidemiological data;

- Also in the introduction, the acronym tricuspid valve (TV) should be placed first in the text as it has already been mentioned previously in the text (it should be also explained in the abstract, albeit easily interpretable);

We fixed the missing acronyms in both the “Introduction” and “Abstract” parts;

- In the methods, it might be useful to summarize the inclusion and exclusion criteria by creating a specific table;

We added a table with summarized inclusion and exclusion criteria in our article;

- In the results, the authors must decide to have a single criterion when using numbers , without alternating numbers and letters (es. 68 vs four);

We adjusted the numbers in the “Results” section (we left only numbers without letters);

- Minor typos (mostly in spacing, could be a layout error during the visualization though).

We corrected spacing errors.